

# Genome-wide identification and expression analysis of the *WRKY* genes in sugar beet (*Beta vulgaris* L.) under alkaline stress

Guo-Qiang Wu, Zhi-Qiang Li, Han Cao and Jin-Long Wang

School of Life Science and Engineering, Lanzhou University of Technology, Lanzhou, China

## ABSTRACT

**Background**. The WRKY transcription factor family plays crucial roles in many aspects of physiological processes and adaption to environment. Although the *WRKY* genes have been widely identified in various plant species, the structure and function of the *WRKY* family in sugar beet (*Beta vulgaris* L.) remains unknown.

**Methods**. In the present study, the *WRKY* genes were identified from the sugar beet genome by bioinformatics. A phylogenetic tree was constructed by MEGA7.0. A distribution map of these genes was displayed by MapInspect 1.0. Furthermore, the exon-intron structure and the conserved motifs were predicted by GSDS 2.0 and MEME 5.0.5, respectively. Additionally, the expression levels of nine selected genes in shoots and roots of sugar beet seedlings exposed to alkaline stress were assayed by qRT-PCR.

**Results**. A total of 58 putative *BvWRKY* genes are identified in the sugar beet genome. The coding sequences of these genes ranged from 558 to 2,307 bp and molecular weights (MWs) varied from 21.3 to 84. The *BvWRKY* genes are clustered into three major groups I, II, and III, with 11, 40, and seven members, based on the primary amino acid sequences. The number of introns in the *BvWRKY* genes ranged from 1 to 5, with a majority of *BvWRKY* (27/58) containing three exons. All the *BvWRKY* genes have one or two conserved WRKY domains and zinc-finger structure. Moreover, the selected *BvWRKY* genes showed a variety of expression patterns in shoots and roots of seedlings under various concentrations of NaHCO₃. Importantly, *BvWRKY10* in shoots and *BvWRKY16* in roots were remarkably up-regulated by alkaline stress. Taken together, our findings extend understandings of the *BvWRKY* genes family and provide useful information for subsequent research on their functions in sugar beet under alkaline stress.

## INTRODUCTION

Plants encounter various abiotic and biotic stresses throughout their life cycle. These stresses can adversely affect the growth and development of plants and/or change the distribution of species (*Prachi et al., 2017*). To cope with these adverse environmental conditions, plants have evolved a variety of mechanisms at the morphological, physiological, cellular

Corresponding author
Guo-Qiang Wu, gqwu@lut.cn, wugq08@126.com

and molecular levels during the process of long-term evolution (*Hasanuzzaman et al., 2013*; *Bechtold & Field, 2018*; *Yang & Guo, 2018*). To date, the responses of abiotic stresses and the regulation of genes have been widely identified in various plant species, such as *Arabidopsis thaliana* (*Kotchoni et al., 2006*), rice (*Oryza sativa*) (*Lee et al., 2005*), tomato (*Lycopersicon esculentum*) (*Sharma et al., 2010*), and wheat (*Triticum aestivum*) (*Zhang et al., 2011*). It is well-documented that several families of genes are particularly related to obvious improvement in abiotic stress tolerance of plants, including the *WRKY*, *NAC*, *MYB*, and *GRF* genes families (*Chinnusamy, Zhu & Zhu, 2006*; *Hennig, 2012*; *Cao et al., 2017*; *Khadiza et al., 2017*).

WRKY proteins are a large family of transcription factors (TFs) involved in various aspects of physiological processes as well as responded to biotic and abiotic stresses (*Jiang et al., 2017*). Since the first *WRKY* gene *SPF1* was cloned from sweetpotato (*Ipomoea batatas*) (*Ishiguro & Nakamura, 1994*), more and more *WRKY* genes have been identified in various plant species, especially some grass species, including barley (*Hordeum vulgare*) (*Mangelsen et al., 2008*), *Brachypodium distachyon* (*Wen et al., 2014*), maize (*Zea mays*) (*Wei et al., 2016*), rice (*Ross, Liu & Shen, 2010*), and wheat (*Zhu et al., 2013*). All the members of the *WRKY* genes have been documented to have one or two conserved WRKY domains, which are composed of approximately 60 amino acids with "WRKYGQK" at N-terminus and a novel zinc-finger motif C-$X_{4-5}$-C-$X_{22-23}$-H-X-H ($C_2H_2$) or C-$X_7$-C-$X_{23}$-H-X-C ($C_2HC$) at C-terminus (*Eulgem et al., 2000*; *Rushton et al., 2010*). Based on the number and characteristics of the conserved WRKY domains, the *WRKY* genes were clustered into three groups I, II, and III (*Eulgem et al., 2000*). There are evidences that the *WRKY* members of group I displayed two WRKY domains with zinc-finger motifs of $C_2H_2$, group II contained only single WRKY domain with a zinc-finger motif of $C_2H_2$, whereas group III had one WRKY domain with a zinc-finger motif of $C_2HC$ (*Rushton et al., 2010*). Additionally, the *WRKYs* of group II can be further classified into five distinct subgroups IIa–e (*Rushton et al., 2010*; *Ulker & Somssich, 2004*). It was also reported that the *WRKY* genes were significantly induced by various environmental factors, such as heat (*Li et al., 2009*), drought (*Jiang, Gang & Yu, 2012*), waterlogging (*Li et al., 2017*), cold (*Luo et al., 2017*), and salt (*Dan et al., 2018*), indicating that the *WRKY* genes play the positive regulatory functions in plants when exposed to adversely stressed conditions. In *Arabidopsis*, overexpression of *TaWRKY1* and *TaWRKY33* from wheat has been shown to activate the expression of several downstream genes related to stress response, increase the germination rate of seeds, and promote the growth of roots in transgenic plants subjected to various stresses (*He et al., 2016*). Compared to wild-type (WT) lines, *TaWRKY33* transgenic *Arabidopsis* lines displayed lower rates of water loss during dehydration (*He et al., 2016*). In rice, *OsWRKY72* conferred more tolerant to salt and drought stresses via ABA signaling (*Song et al., 2010*). *OsWRKY74* overexpressing rice exhibited greater accumulation of iron (Fe) and up-regulation of the cold-responsive genes compared with WT plants in P-deficient conditions (*Dai, Wang & Zhang, 2016*). Under saline condition, the activities of catalase (CAT), peroxidase (POD), and superoxide dismutase (SOD) in *Chrysanthemum* overexpressed *DgWRKY5* gene were significantly higher than those of WT plants, whilst the accumulation of $H_2O_2$, $O^{2-}$, and malondialdehyde (MDA) were significantly lower than

those of WT plants (*Liang et al., 2017*). Moreover, the expression levels of genes related stress such as *DgCAT*, *DgAPX*, *DgNCED3A*, *DgCuZnSOD*, *DgNCED3B*, *DgCSD1*, *DgCSD2*, and *DgP5CS* were remarkably higher in transgenic *Chrysanthemum* plants than those in WT plants, indicating that DgWRKY5 might be a positive regulatory factor in response to salt stress (*Liang et al., 2017*). These results documented that the *WRKY* genes may provide valuable insights into abiotic stress tolerance mechanisms in plants.

Sugar beet ($2n = 18$, *Beta vulgaris* L.), belonging to the order of Caryophyalles, is a major sugar crop worldwide, which provides approximately 30% of the world's sugar production (*Liu et al., 2010*). In China, this crop is cultivated in the arid and semi-arid regions of Northern China (*Wu et al., 2013*). The whole-genome sequence of sugar beet was completed and released in 2014, and a total of 359.14 Mb of sequence data were assembled with 27,421 protein-coding genes predicted (*Dohm et al., 2014*). Our previous studies indicated that the addition of 50 mM NaCl in the growth medium can stimulate the growth of plants and mitigate the damage caused by osmotic stress in sugar beet (*Wu et al., 2015*). Recently, our results showed that $Na^+$ concentration in both roots and shoots displayed a sharply increased trend with the increase of $NaCHO_3$ concentrations, whereas $K^+$ concentration maintained a relatively stable level under either low- or high-alkaline conditions (Dataset S1). These results implied that the maintenance of $K^+$ and $Na^+$ homeostasis might be an important strategy for sugar beet adapting to alkaline stress. However, the *WRKY* family genes and their regulated expression in sugar beet exposed to alkaline stress still remain unknown.

Here, we proposed a hypothesis that the *WRKY* genes play a positive regulatory function in response to alkaline stress in sugar beet. To test this hypothesis, in this study, firstly, a total of 58 *BvWRKY* genes were identified in the sugar beet genome, and the phylogenetic relationship, chromosome distribution, genes structure and conserved motifs were analyzed; Secondly, the expression patterns of the *BvWRKY* genes in roots and shoots of sugar beet seedlings exposed to different concentrations (15–100 mM) of $NaCHO_3$ were determined by qRT-PCR. Our findings extend understandings of the *BvWRKY* family genes and provide useful information for subsequent research on their functions in sugar beet under alkaline stress.

## MATERIALS AND METHODS

### Identification and distribution of the *WRKY* genes in sugar beet

To identify the sugar beet *WRKY* family genes, protein sequences of the *WRKY* genes in *Arabidopsis* were downloaded from the *Arabidopsis* Information Resource (TAIR) (https://www.arabidopsis.org/), and used queries in performing on Basic Local Alignment Search Tool of protein (BLASTp) searches with default algorithm parameters on the *Beta vulgaris* Resource (http://bvseq.boku.ac.at/) (*Dohm et al., 2014*) and NCBI sugar beet genome data (https://www.ncbi.nlm.nih.gov/nuccore/?term=Beta+vulgaris+subsp.+vulgaris). The hidden Markov Model (HMM) profiles of the conserved WRKYGQK domain PF03106 were obtained from Pfam 32.0 (http://pfam.xfam.org/) and the HMM 3.1 search tool (https://www.ebi.ac.uk/Tools/hmmer/) was used to identify all WRKY proteins in sugar

beet (*Potter et al., 2018*). Candidate WRKY proteins were manually further validate (*Wang et al., 2014*) via searching for WRKY domains in the amino acid sequences using online tool SWISS-MODEL (https://swissmodel.expasy.org/) (*Benkert, Biasini & Schwede, 2011*). Redundant and incomplete residual sequences were removed from protein sequences with complete WRKYGQK domains by using DNAMAN 6.0 (*Yue et al., 2019*).

The isoelectric points (pIs) and theoretical molecular weights (MWs) of each BvWRKY protein were predicted online at https://web.expasy.org/protparam/ (*Gasteiger et al., 2005*). Based on the position information from the sugar beet genome database, the *BvWRKY* genes were plotted on the chromosomes and the distribution map of *BvWRKY* genes was displayed by MapInspect 1.0 (https://mapinspect.software.informer.com/). Tandem and segmental duplication events of *BvWRKY* genes were analyzed according to the methods as described by *Hu & Liu (2012)*.

## Phylogenetic analysis, gene structure and conserved motifs distribution

The amino acid sequences of *BvWRKY* genes were aligned with those of *AtWRKYs* from *Arabidopsis* (Table S1), by using the Clustal W 2.0 (http://www.clustal.org/clustal2/) (*Larkin et al., 2007*). Phylogenetic tree was constructed by MEGA 7.0 (https://www.megasoftware.net/) using the neighbor-joining (NJ) algorithm with 1,000 bootstraps. The exon-intron structures of the *BvWRKY* genes were predicated via Gene Structure Display Server (GSDS 2.0, http://gsds.cbi.pku.edu.cn/) according to the alignments of their coding sequences (CDS) with their corresponding genomic sequences (*Hu et al., 2015*). Multiple Em for Motif Elicitation (MEME 5.0.5, http://meme-suite.org/tools/meme) was used to predict the conserved motifs of the *BvWRKY* genes (*Bailey & Elkan, 1994*), and the parameters used in this study were set as follows: maximum number of different motifs is 10, other default parameters.

## Gene ontology analysis

Gene ontology (GO) terms of *BvWRKYs* were predicted by using online tool DAVID Bioinformatics Resources 6.8 (https://david.ncifcrf.gov/home.jsp) (*Huang, Sherman & Lempicki, 2009*). The analysis was based mainly on three aspects: cellular component, biological process, and molecular function.

## Three-dimensional structure analysis of BvWRKY

The three-dimensional (3D) structures of nine selected proteins, namely BvWRKY3, -10, -16, -22, -41, -42, -44, -47, and -51, were predicted by online service I-TASSER (https://zhanglab.ccmb.med.umich.edu/I-TASSER/) (*Yang et al., 2015*). In order to identify structurally similar templates in the Protein Data Bank (PDB) database (*Berman et al., 2000*), the query sequences were subjected to multiple rounds of threading using the LOMETS method (*Zheng et al., 2019*).

## Plant materials, growth conditions and alkaline stress treatments

Seeds of sugar beet (*B. vulgaris* L.) cultivar "Gantang7" were kindly provided by Wuwei Sannong Seed Technology Co., Ltd., Gansu, China, in May 2018. Seeds were surface

sterilized for 1 min in 75% ethanol ($v/v$) and rinsed three times with sterilized distilled water, soaked in sterilized water for 24 h (*Wu et al., 2013*; *Wu et al., 2015*), and then planted in plastic containers filled with vermiculite and watered with the modified Hoagland nutrient solution including 2 mM $KNO_3$, 1 mM $NH_4H_2PO_4$, 0.5 mM $Ca(NO_3)_2$, 18 mM $MnCl_2 \cdot 4H_2O$, 1.6 mM $ZnSO_4 \cdot 7H_2O$, 0.6 mM $CuSO_4 \cdot 5H_2O$, 0.5 mM $MgSO_4$, 60 mM Fe-Citrate, 92 mM $H_3BO_3$, and 0.7 mM $(NH_4)_6Mo_7O_{24} \cdot 4H_2O$. All the seedlings were grown in the same growth chamber at temperature of 25/20 °C (day/night), daily photoperiod of 16/8 h (day/night), relative humidity of 65–75%, and light density of 550–600 mmol $\cdot$ m$^{-2}$ $\cdot$ s$^{-1}$ during the photoperiod.

Four-week-old seedlings with uniform size were subjected with the modified Hoagland nutrient solution supplemented with additional 0, 15, 25, 50, and 100 mM $NaHCO_3$. Shoot and root tissues were collected at 72 h after $NaHCO_3$ treatments, respectively. Samples of shoots and roots were immediately frozen in liquid nitrogen and stored at −80 °C until RNA extraction.

## qRT-PCR analysis

To validate functions of *BvWRKYs* in response to alkaline stress, nine genes, including 5 genes in group I (*BvWRKY10, -22, -41, -44,* and *-51*) and 4 genes in group II (*BvWRKY3, -16, -42,* and *-47*), were selected according to homologous genes in rice and *Arabidopsis* (*Wu, 2005*), and their expression patterns were analyzed by using qRT-PCR. The total RNA was isolated from roots or shoots of sugar beet using UNlQ-10 Column Trizol Total RNA Isolation kit (Sangon, Shanghai, China) according to the manufacturer's procedure. The cDNAs were synthesized using a PrimeScript$^{TM}$ Real-Time (RT) Master Mix kit (Takara, Dalian, China) according to the manufacturer's instruction. qRT-PCR with a TB Green$^{TM}$ Premix Ex Taq$^{TM}$ II kit (Takara, Dalian, China) was performed using a MA-6000 RT-PCR System (Molarray, Suzhou, China). The qRT-PCR reaction conditions were as following: 95 °C for 30 s, followed by 40 cycles of 95 °C for 5 s, and 60 °C for 1 min. The expression levels of each gene were determined according to the $2^{-\Delta\Delta}CT$ method (*Livak & Schmittgen, 2001*). Three biological replicates were used. The *BvACTIN* gene was used as the internal control. All the primers of nine *BvWRKY* genes used for qRT-PCR are listed in Table 1.

## Phenotype analysis

To further determine the effects of alkaline stress on phenotype of sugar beet, four-week-old seedlings were treated with the above alkaline concentrations for 7 d. At the end of treatments, samples of shoots and roots were separated and collected, respectively. Fresh weights (FW) of shoots and roots were assayed immediately. Dry weights (DW) of shoots and roots samples were determined after oven drying at 80 °C until constant weight. Each treatment consisted of eight biological replicates. Two seedlings were pooled in each replicate. Data were analyzed by using one-way analysis of variance (ANOVA) at a significant level of $P < 0.05$.

**Table 1** Sequences of primers used in qRT-PCR.

| No. | Gene name | Forward primer sequence (5′-3′) | Reverse primer sequence (5′-3′) |
| --- | --- | --- | --- |
| 1 | *BvACTIN* | ACTGGTATTGTGCTTGACTC | ATGAGATAATCAGTGAGATC |
| 2 | *BvWRKY3* | CCTCATGGATGAACTACAAAACGTCG | ATCAACGGCATCCGAAACGTTAATC |
| 3 | *BvWRKY10* | CTCCAGATGATGTTCCAAGGACAC | GGCACAGCAAGAAAGAGAAGTG |
| 4 | *BvWRKY16* | CGGCTACCACTAGACTTAGCTCCT | GTCTTTAAGCTCATCTTGTGACGTGC |
| 5 | *BvWRKY22* | CTCAACCTAATCGCCGACTTC | ATTAAATGGAGGCACGCGGT |
| 6 | *BvWRKY41* | CGGAAAATCTCACAACTCCCTCTTCT | TTCGGAGAAGAAACTCGAGACCAG |
| 7 | *BvWRKY42* | GGAGACCGAGATCAGTGGTTCTTC | TACTTCTCCCATCTTTGCTTTGGC |
| 8 | *BvWRKY44* | GGCTCCTTCTTCACTTTCTGTCTC | CCACCAAATGCTCCTACAGTTG |
| 9 | *BvWRKY47* | CTACCTCAAGCTAGCATGGAAGCAA | TCTTAGGAGATGATATGGAGGCGGC |
| 10 | *BvWRKY51* | GCAGTGATTGTAGCTCCTAAGGTT | ATGGTTTCTCAGGGACAACAGA |

## RESULTS

### Identification and characterization of the *BvWRKY* genes

A total of 64 transcripts in the sugar beet genome sequences were identified as possible candidate members of the WRKY family. After removing the incomplete sequences without the conserved WRKY domain, a total of 58 sequences were eventually identified as the putative sugar beet *WRKY* genes and named as *BvWRKY1* to *BvWRKY58* according to their order in the sugar beet genomic sequence (Table S2). The sequence analysis of the *BvWRKY* genes showed that CDS ranges from 558 (*BvWRKY24*) to 2,307 bp (*BvWRKY18*) and predicted proteins ranged from 185 to 768 amino acids (aa) in length with an average of approximately 387 aa. The MWs varied from 21.3 (*BvWRKY24*) to 84.2 kDa (*BvWRKY18*). The pIs ranged from 5.4 (*BvWRKY7*) to 9.9 (*BvWRKY19*), with 22 pIs >7 and the remaining pIs ≤7 (Table S2).

### Classification and phylogenetic analysis of the *BvWRKY* genes

To determine the structural features of each BvWRKY protein, multiple sequences alignments were performed by using DNAMAN 6.0 (Fig. S1). Of the 58 BvWRKY proteins identified, the majority (47, 81%) had one conserved WRKYGQK domain, and remaining 11 (19%) contained two conserved WRKYGQK domains (Fig. S1 and Table S2). Although the WRKYGQK domain was absolutely conserved in the WRKY families, two proteins (BvWRKY7 and -8) differed at one residue, with a glutamine (Q) being replaced by a lysine (K) residue; similar change was also observed in other plant species such as barley (*Mangelsen et al., 2008*), tomato (*Huang et al., 2012*) and sesame (*Sesamum indicum*) (*Li et al., 2017*). Furthermore, all the BvWRKY proteins contained the C-X$_{4-7}$-C-X$_{22-23}$-H motif that forms zinc-finger structures of C$_2$HC or C$_2$H$_2$.

To explore the phylogenetic and evolutionary relationship of the *WRKY* genes in sugar beet and group them with the established subfamilies, we investigated 118 amino acid sequences containing the conserved WRKYGQK domain. These sequences consisted of 58 sequences from sugar beet and 60 sequences from *Arabidopsis* (Table S1). An unrooted NJ phylogenetic tree was constructed according to multiple alignments of the predicted amino acid sequences by MEGA 7.0 (Fig. 1). According to WRKY classification in *Arabidopsis*
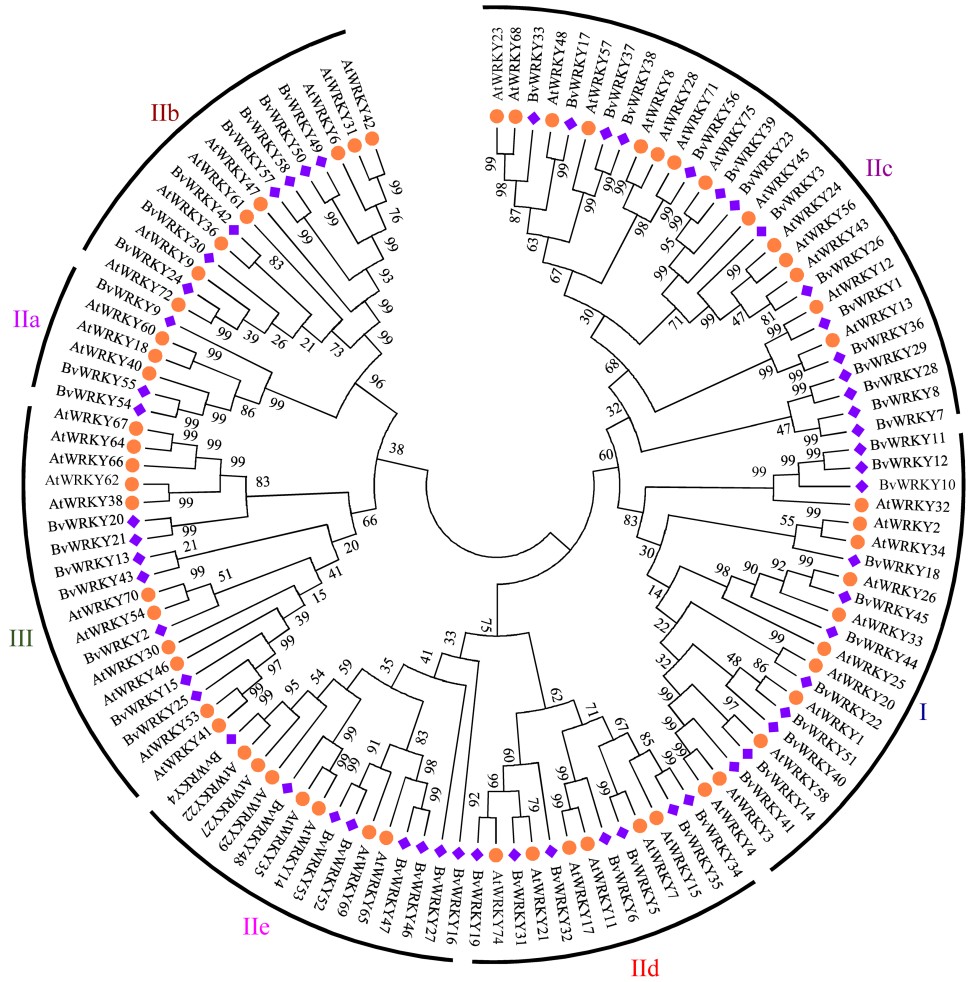

**Figure 1** Phylogenetic tree of *WRKY* genes in sugar beet (*Beta vulgaris, Bv*) and *Arabidopsis thaliana* (*At*). The predicted proteins sequences of 58 *BvWRKYs* and 60 *AtWRKYs* were aligned by the Clustal W and the phylogenetic tree was constructed using the MEGA7.0 by the NJ method with 1,000 bootstrap replicates. The *WRKY* genes were clustered into three major groups. *AtWRKYs* and *BvWRKYs* are represented in orange circles and blue squares, respectively. Details of sequences of *BvWRKYs* and *AtWRKYs* are listed in Table S1.

(*Eulgem et al., 2000*), the 58 *BvWRKY* genes were clustered into three major groups I, II, and III. There are 11 members in the *BvWRKY* group I, 40 members in group II, and seven members in group III. Moreover, the *BvWRKYs* in group II were subdivided into five subgroups IIa, b, c, d, and e, containing three, seven, 15, seven, and eight members, respectively (Fig. 1 and Table S2).

## Chromosomal distribution of the *BvWRKY* genes and their genomic duplication

To examine the genomic distribution of the *WRKY* genes, the *BvWRKY* genes were mapped on their corresponding chromosome by searching the released genomic database of sugar beet. The results showed that 55 *BvWRKY* genes are unevenly distributed on nine

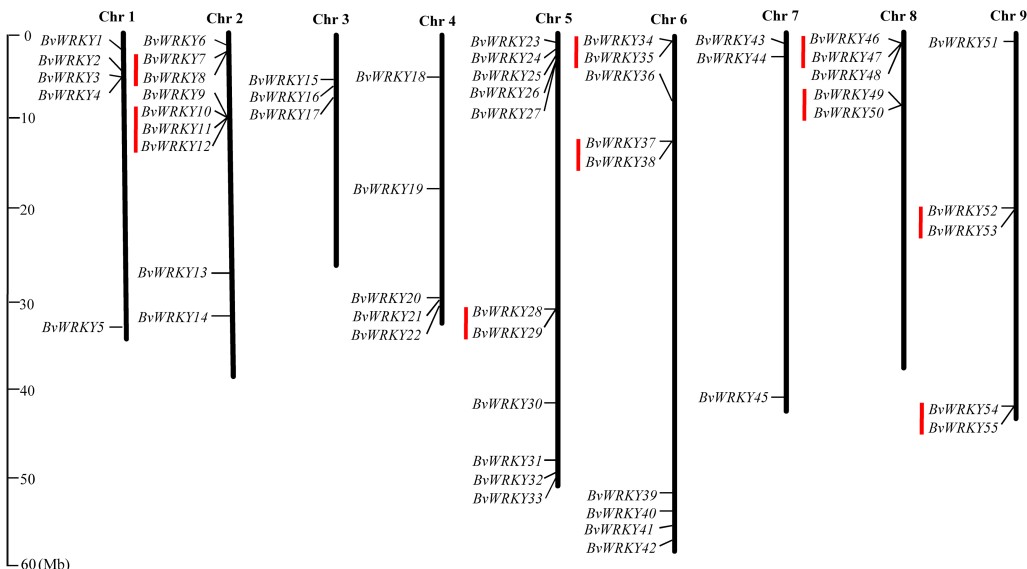

**Figure 2 Distribution of the *BvWRKY* genes on the sugar beetchromosomes.** The chromosome number is indicated at the top of each chromosome. The scale of the genome size is given on the left. Red lines represent tandem gene duplications.

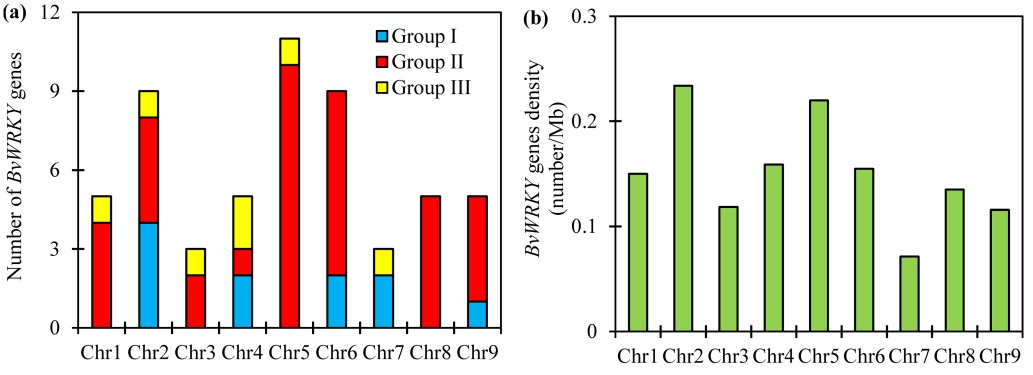

**Figure 3 Unevenly chromosomal distributionof the *BvWRKY* genes.** (A) Number of *BvWRKY* genes in each chromosome. (B) The *BvWRKY* genes density per chromosome.

sugar beet chromosomes (Fig. 2), and the number on each chromosome is not necessarily correlated with its length. Chr 2, 5, and 6 had relatively more *BvWRKY* genes, with nine, 11, and nine genes, respectively. Chr 3 and 7 contained relatively fewer *BvWRKY* genes, with only three genes, respectively (Fig. 3A). The other three genes (*BvWRKY56, -57,* and *-58*) were mapped onto unassembled scaffolds based on the current database (Table S2). The *BvWRKY* genes density per chromosome ranged from 0.071/Mb to 0.234/Mb (Fig. 3B). Averagely, one *BvWRKY* gene was present every 6.53 Mb. Several chromosomes have higher densities of *BvWRKY* genes compared to others. Chr 2 has the highest density of *BvWRKY* genes, whereas Chr 7 has the lowest density (Fig. 3B).

Two or more homologous genes within a 100 Kb range distance were defined as tandem duplicates. Nine tandem duplication regions clustered with 19 *BvWRKY* genes, including three genes in group I, two genes in group IIa, two genes in group IIb, 6 genes in group IIc, two genes in group IId, and four genes in group IIe, are identified on Chr 2 (*BvWRKY7* and *-8*, and *BvWRKY 10, -11, -12*), Chr 5 (*BvWRKY28, -29*), Chr 6 (*BvWRKY34, -35* and *BvWRKY37, -38*), Chr 8 (*BvWRKY46, -47* and *BvWRKY49, -50*), and Chr 9 (*BvWRKY52, -53* and *BvWRKY54, -55*), respectively (Fig. 2). The segmental duplications were further analyzed on 100 kb DNA segments flanking each *BvWRKY* gene and no gene was found to be attributed to segmental duplication. Therefore, it is speculated that tandem duplication most likely played an important role in the observed gene expansion of sugar beet *WRKY* genes.

## Conserved motifs and structure of the *BvWRKY* genes

To further examine the structural characteristics of the *WRKY* genes in sugar beet, the conserved motifs of BvWRKY proteins were predicted by using the MEME 5.0.5 and further annotated by using InterPro Scan 5.0 (*Jones et al., 2014*). A total of 10 putatively conserved motifs were identified in the BvWRKY proteins (Fig. 4). The identified amino acids length of BvWRKY motifs varied from 15 to 50. Detailed sequences of BvWRKY motifs are shown in Fig. S2. The number of motifs was different in those proteins, varying from two to six. The results showed that motif 1 and *-3* were annotated as WRKY domains, while motif 2 was found in the zinc-finger domain. Notably, similar motif compositions were observed in the same group of BvWRKY proteins. Motif 3 and *-6* were found in group I and motif 7 in the IIa and IIb subgroups (Fig. 4).

To determine the structural diversity of the *BvWRKY* genes, the distribution of intron-exon was analyzed and compared. The number of exons in *BvWRKYs* ranged from 2 (*BvWRKY3, -6, -23, -24, -26,* and *-39*) to 6 (*BvWRKY10, -11, -12, -18, -22, -40, -49, -50, -57,* and *-58*). It was found that 53.4% of *BvWRKY* genes contained the typical splicing of three exons and two introns (Fig. 5). For members of group II, the IIa subgroup contained 4 exons, subgroup IIb contained 2–6 exons, the IIc and IId subgroups contained 3 exons, and subgroup IIc contained 2–4 exons (Fig. 5). These results suggested that *BvWRKY* genes displayed the diversity of intron/exon structure.

## GO analysis of *BvWRKY* genes

To reveal the functional classifications of the *BvWRKY* genes, GO terms were predicted by using online tool DAVID 6.8. The results showed that the *BvWRKY* genes were categorized into seven functional groups under main three categories: cellular component, biological process and molecular function (Fig. 6 and Table S2). More genes were classified under "sequence-specific DNA binding (GO:0043565)" and "transcription factor activity (GO:0003700)" than the five other functional groups, such as "transcription regulatory region DNA binding (GO:0031347)", "regulation of defense response (GO:0031347)", "positive regulation of transcription (GO:0045893)", "nucleus (GO:0005636)" and cytoplasm "GO: 0005737" (Fig. 6 and Table S2).

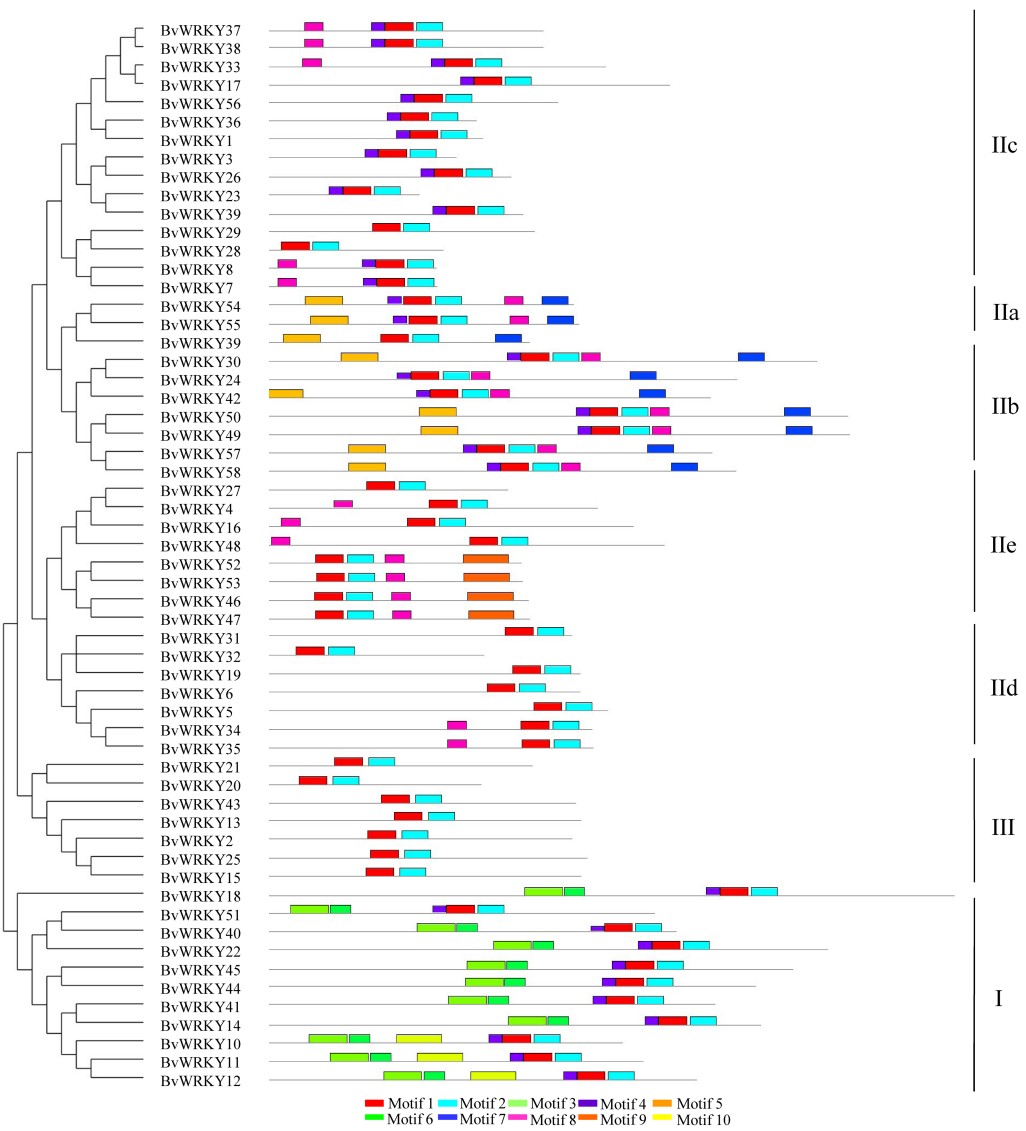

**Figure 4  BvWRKY proteins motifs identified by MEME using the complete amino acid sequences.**
Combined *p*-values are indicated and different motifs are shown by different colors and numbered from 1
to 10. Detailed information of BvWRKY motifs is listed in Fig. S2.

## 3D structure prediction of BvWRKY proteins

To understand the structural characteristics of BvWRKY proteins, 3D structures were
constructed according to the similar template obtained from PDB using I-TASSER. The
modeled structures for the selected BvWRKY proteins contained 3 (BvWRKY3) to 58
(BvWRKY42) $\alpha$-helices and 19 (BvWRKY42) to 38 (BvWRKY10) $\beta$-strands (Fig. 7 and
Fig. S3). All the predicted BvWRKY models showed a C-score range from −2.57 to −4.44
(Table S3), indicating that the predicted proteins are constructed with high accuracy.

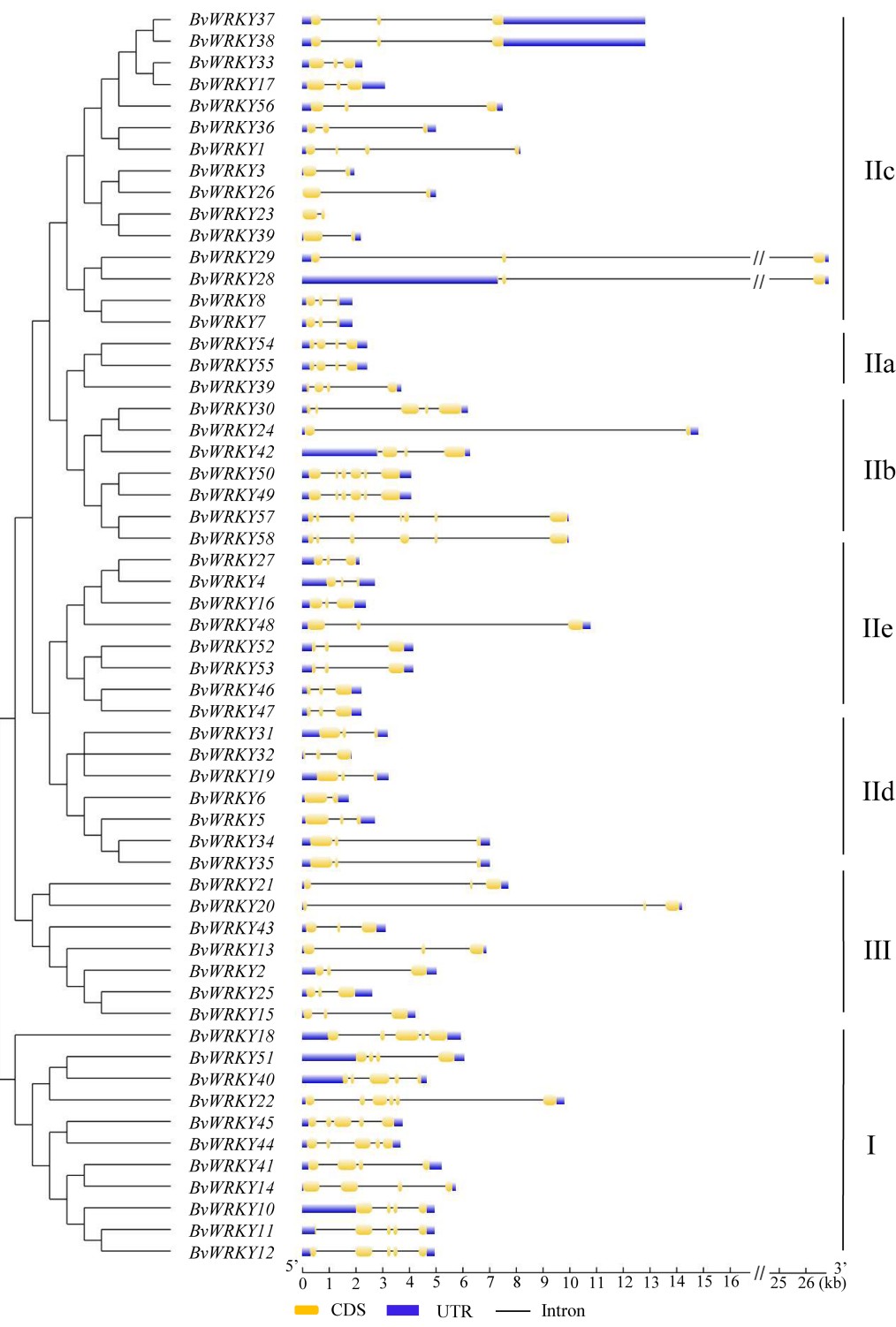

**Figure 5 The exon-intron structures of the *BvWRKY* genes.** Exon-intron structure analyses of the *BvWRKY* genes were performed by using the online tool GSDS. Lengths of exons and introns of each *BvWRKY* gene were exhibited proportionally. Introns are represented by black lines. Exons are represented by yellow boxes. UTR are represented by blue boxes. The scale of genes length is given at the bottom. CDS, coding equence; UTR, untranslated region.

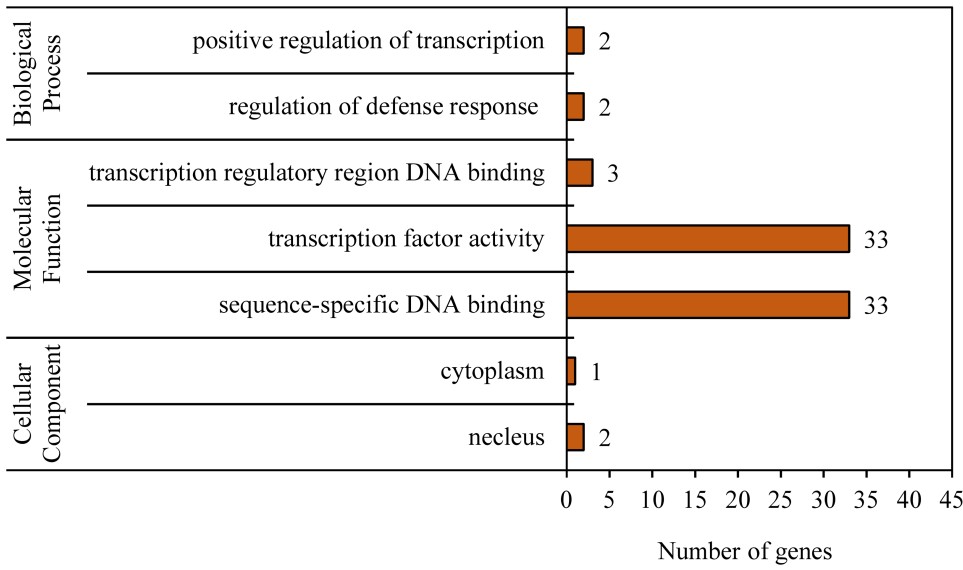

**Figure 6   Gene ontology (GO) analysis of the *BvWRKY* genes.** The lengths of the rectangular columns indicate the number of genes that participated in the corresponding classification. Categories pertaining to cellular component, molecular function and biological process were defined by GO classification.

## Expression analysis of 9 *BvWRKY* genes in response to alkaline stress

To further confirm if the expression of *BvWRKY* genes was induced by alkaline stresses, 9 genes, namely *BvWRKY3*, *-10*, *-16*, *-22*, *-41*, *-42*, *-44*, *-47*, and *-51*, were selected according to homologous genes in rice and *Arabidopsis* which play the vital roles in abiotic stresses (*Wu, 2005*), and qRT-PCRs were performed to analyze their expression patterns in response to different concentrations of $NaCHO_3$. The results showed that the expression patterns of the most detected *BvWRKY* genes differed in response to alkaline stresses (Fig. 8 and Table S4). With the increase of $NaCHO_3$ concentrations, the transcript abundances of three genes (*BvWRKY10*, *-42*, and *-47*) in shoots gradually increased at 15 and 25 mM, then reached a maximum value at 50 mM which are 3.2-, 1.7-, and 1.6-fold of those at control (0 mM), respectively, and then either maintained a higher level (*BvWRKY10* and *-42*) or showed a significant reduction (*BvWRKY47*) at 100 mM (Figs. 8B, 8F and 8H). Additionally, the expression levels of *BvWRKY3* or *BvWRKY51* in shoots were unchanged or repressed by alkaline stresses compared with control. It is also showed that the expression levels of both *BvWRK16* and *-42* in roots displayed a significant enhancement at 25 mM, then reached a peak value at 50 mM which are 12.6- and 1.4-fold higher than those at control, respectively, and then exhibited a reduction at 100 mM (Figs. 8C and 8F). Interestingly, two genes (*BvWRKY22* and *-41*) showed a down-, up- and then down-regulation pattern in shoots (Figs. 8D and 8E), whereas other genes (*BvWRKY44* and *-51*) displayed an up-, down- and then up-regulation pattern in roots with the increase of alkaline concentrations (Figs. 8G and 8I). These results demonstrated that sugar beet *WRKYs* expression patterns vary in response to alkaline stresses.

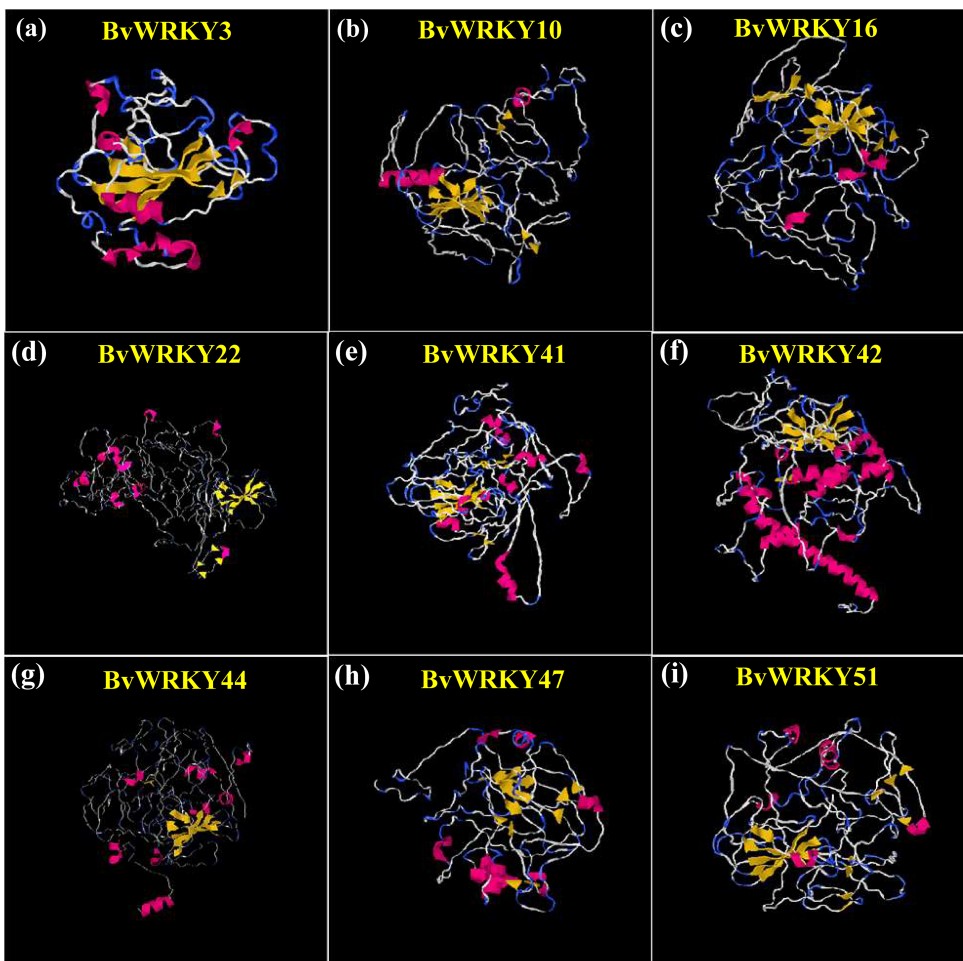

**Figure 7  Predicted three-dimensional (3D) structure of nine selected BvWRKY proteins.** (A) BvWRKY3; (B) BvWRKY10; (C) BvWRKY16; (D) BvWRKY22; (E) BvWRKY41; (F) BvWRKY42; (G) BvWRKY44; (H) BvWRKY47; (I) BvWRKY51. The models were gained by I-TASSER. $\alpha$-helices, $\beta$-strands and random coils are shown in red, yellow and blue, respectively. The best Protein Data Bank (PDB) structural analog for each protein is shown in Table S3. Details of secondary structure of BwWRKY proteins are represented in Fig. S3.

## Phenotype analysis of sugar beet under alkaline stress

To further understand alkaline tolerance of sugar beet, shoots and roots of seedlings were harvested at 7 d after different concentrations of $NaHCO_3$ treatments. The results showed that additional 15 mM $NaHCO_3$ significantly increased fresh weight (FW), but not dry weight (DW) in shoots compared with control (0 mM), while concentrations of 25–100 mM had no significant effects on both FW and DW in shoots. Compared to control, all concentrations of $NaHCO_3$ did not have any effects on either FW or DW in roots (Fig. S4 and Table S5). These results suggested that sugar beet displays strongly tolerant to alkaline stress.

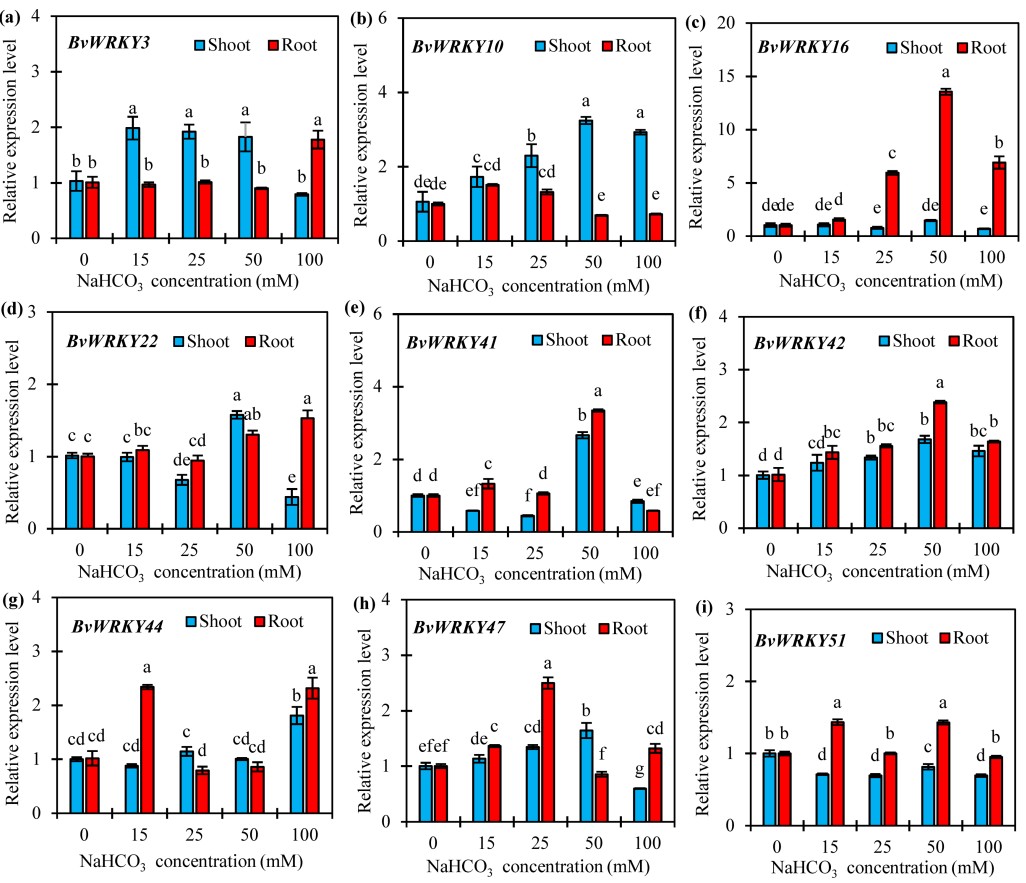

**Figure 8** **Relative expression levels of nine selected *BvWRKY* genes in shoot and root of sugar beet seedlings exposed to 0, 15, 25, 50, and 100 mM NaHCO₃ for 72 h.** (A) *BvWRKY3*; (B) *BvWRKY10*; (C) *BvWRKY16*; (D) *BvWRKY22*; (E) *BvWRKY41*; (F) *BvWRKY42*; (G) *BvWRKY44*; (H) *BvWRKY47*; (I) *BvWRKY51*. Expression of the *BvWRKY* genes normalized to those of *BvACTIN* and shown relative to the expression at 0 mM NaHCO₃. The $2^{-\Delta\Delta Ct}$ method was used to calculate the expression levels of target genes at different treatments. Experiments were repeated at least three times. Values are means ±SE and bars indicate SE ($n = 3$). Columns with different letters indicate significant differences at $P < 0.05$ (Duncan's test).

## DISCUSSION

The *WRKY* family has been widely identified in various organisms, including spike mosses, single-celled green algae, slime molds and protozoa (*Rushton et al., 2010*). In monocots and dicots, including soybean (*Glycine max*) (*Luo et al., 2013*), wheat (*Qin, Tian & Liu, 2015*), rice (*Dai, Wang & Zhang, 2016*), and cotton (*Gossypium hirsutum*) (*Liu et al., 2016*), an especially large number of *WRKY* genes have been documented to have various functions in recent years. In the present study, the *WRKY* genes were firstly identified from whole-genome sequences of sugar beet.

To date, whole genomes of many plants have been sequenced and a large of the *WRKY* genes have been identified in various plant species (*Wu, 2005*; *Wei et al., 2012*; *Dou et al., 2014*; *Yu et al., 2016*; *Yue et al., 2016*; *Jing et al., 2017*). Completion of the sugar beet

genome makes it possible to analyze the *WRKY* genes at the whole genome level (*Dohm et al., 2014*). In the present study, a total of 58 putative *BvWRKY* genes are identified in the sugar beet genome (Table S2). It was found that there are 32 *WRKY* genes in broomcorn millet (*Panicum miliaceum*) (*Yue et al., 2016*), 71 in sesame (*Li et al., 2017*), 85 in cassava (*Manihot esculenta*) (*Wei et al., 2016*), 88 in common bean (*Phaseolus vulgaris*) (*Jing et al., 2017*), 100 in rice (*Wu, 2005*), 103 in *Aegilops tauschii* (*Ma et al., 2014*), 116 in *Gossypium raimondii* (*Dou et al., 2014*), and 136 in maize (*Wei et al., 2012*). These findings suggested that there are large differences in number of the *WRKY* genes families among various plant species.

These are consistent with the classification of the *WRKY* family genes in many plant species such as *Arabidopsis* (*Eulgem et al., 2000*), maize (*Wei et al., 2012*), *Populus* (*Jiang et al., 2014*), cassava (*Wei et al., 2016*), and peach (*Prunus persica*) (*Chen et al., 2016*). All the *WRKY* genes can be classified into three distinct clusters: group I, II, and III depending on the number of conserved WRKY regions and the pattern of zinc-finger motif (*Eulgem et al., 2000*; *Wei et al., 2012*; *Jiang et al., 2014*). There are evidences that genes of group I included double conserved WRKY domains, which can interact with the W-box "TTGACC/T" core motif to activate downstream genes, and $C_2H_2$ zinc-finger motif; group II only possessed single WRKY domain and shared the same zinc-finger motif as group I; whereas group III had one conserved WRKY domain and $C_2HC$ zinc-finger motif (*Eulgem et al., 2000*; *Rushton et al., 2010*). In this study, 11, 40, and 7 *BvWRKY* genes have been classified into groups I, II, and III based on *Eulgem et al. (2000)* and by using the WRKY proteins in *Arabidopsis* as references, respectively (Fig. 1 and Table S2). Although the WRKYGQK domain was considered to be highly conserved in the WRKY family, tow proteins (BvWRKY7 and BvWRKY8) in group II differed at one residue, with a glutamine being changed into a lysine (K) residue (Fig. S1). This change has been found in many plant species such as barley (*Mangelsen et al., 2008*) and tomato (*Huang et al., 2012*). Furthermore, previous studies have documented that group III was the largest group of *WRKY* genes families in rice and broomcorn millet, which accounted for 38% and 50% (*Ross, Liu & Shen, 2010*; *Yue et al., 2016*), while in *Arabidopsis* and sesame, group II was the largest group, accounting for 24% and 68% (*Eulgem et al., 2000*; *Li et al., 2017*), respectively. In the present study, group II has been also found to be the largest group of *WRKY* genes family in sugar beet, accounting for 69% of all the *BvWRKY* genes, which are consistent with the results of *Arabidopsis* and sesame but different from rice and broomcorn millet. Moreover, group II can be divided into five distinct subgroups IIa, b, c, d, and e, according to the amino acid sequences outside the WRKY domain (Fig. 1 and Table S2). Subgroup IIc is also found to be the largest subgroup, accounting for 37.5% of all the genes of group II (Table S2), which is in accordance with the results reported in soybean (*Luo et al., 2013*), *Arabidopsis* (*Qin, Tian & Liu, 2015*), rice (*Dai, Wang & Zhang, 2016*), and cotton (*Liu et al., 2016*). Additionally, there are closely evolutionary relationships between group IIa and IIb, and between group IId and IIe (Fig. 1), respectively, which appear to make up monophyletic clades. Furthermore, according to the distance of phylogenetic relationship, three groups of the *BvWRKY* genes can be clustered in four major lineages: group IIc + I, group IId + IIe, group III, and group IIa + IIb, respectively (Fig. 1). Similar

lineages were also found in the *MdWRKY* genes families from apple (*Lui et al., 2017*). These results further confirmed that the WRKY family genes are highly conserved family in different plant species.

Gene duplication events played critical roles in rapid expansion and the evolution of genes families (*Cannon et al., 2004*). It is well-known that genes within a single genome are divided into five distinct classes: singletons, dispersed-, proximal-, tandem- and segmental/whole genome duplication (WGD)-duplicates, respectively, according to the copy number of genes and the distribution of genome (*Wang et al., 2012*). It was documented that duplication events can lead to a clustered occurrence of family members via tandem amplification, or a scattered occurrence via segmental duplication of chromosomal regions (*Grassi, Lanave & Saccone, 2008*). In the present study, 32.8% (19/58) of the *BvWRKY* genes likely evolved from tandem repeats (Fig. 2). Tandem gene replications of *WRKYs* have been found in *Arabidopsis* (*Cannon et al., 2004*), rice (*Ross, Liu & Shen, 2010*), cucumber (*Cucumis sativus*) (*Ling et al., 2011*), and soybean (*Yu et al., 2016*). However, it was found that no gene was attributed to segmental duplication in sugar beet *WRKY* genes. Therefore, it is proposed that tandem duplication might be major attributor in the expansion of the *BvWRKY* genes in sugar beet.

The number of motifs in *BvWRKYs* ranges from 2 to 6, and the length of motifs varied from 15 to 50 amino acids (Fig. 4 and Fig. S2). In addition, 4 motifs (motif 1, -2, -3, and -6) are found in the WRKY DNA-binding domain. Similar motifs compositions were reported in *SiWRKYs* from sesame (*Li et al., 2017*). The other 6 motifs are found to be located outside in the WRKY domain. It is clear that motif 1 and -2 are shared by all the *BvWRKY* genes, while motif 3 and -6 were shared by 11 genes, viz *BvWRKY10*, *-11*, *-12*, *-14*, *-18*, *-22*, *-40*, *-41*, *-44*, *-45*, and *-51* (Fig. 4), which belonged to members of group I (Table S2). Importantly, motif 5, -7, -8, and -9 were shared by members of group II, and motif 10 was shared by group I. It is clear that members of *WRKY* family in the same cluster commonly shared similar motif compositions in sugar beet genome.

The structural diversity of exon/intron, an important part in the evolution of gene families, provides an additional evidence supporting phylogenetic classification (*Bleecker, 2003*; *Wang et al., 2014*). In the present study, the number of introns found in the *BvWRKY* genes ranges from 1 to 5, with an average of 2.79 introns per *BvWRKY* gene, so each sequence of *BvWRKY* was divided into many segments by introns. Similarly, all the *WRKY* genes in both cassava and peach have 1–5 introns (*Wei et al., 2016*; *Chen et al., 2016*). However, the *SiWRKY* genes in sesame have 1–11 introns (*Li et al., 2017*). These results implied that the *WRKY* genes showed the diversity of structures in various plant species. Moreover, the largest fraction of *BvWRKYs* (27, 46.6%) have 2 introns (Fig. 5), which is common in other plant species, including cassava (42 of 85) (*Wei et al., 2016*), peach (29 of 58) (*Chen et al., 2016*), and sesame (33 of 71) (*Li et al., 2017*).

Alkaline is one of the most serious abiotic stresses that limits plant growth and crop productivity in the arid and semi-arid regions of Northern China (*Guo, Shi & Wang, 2010*). However, few alkaline tolerant genes have been identified in sugar beet. Recent researches have demonstrated that the *WRKY* genes are involved in various abiotic stresses, especially salt stress (*Zhou et al., 2015*; *Liang et al., 2017*; *Lui et al., 2017*;

*Wang et al., 2017*; *Wu et al., 2017*). There are evidences that a majority of *WRKYs* were remarkably up-regulated only by salinity in various plant species, such as cucumber (*Ling et al., 2011*) and tomato (*Huang et al., 2012*). In *B. distachyon*, however, the most *BdWRKY* genes were significantly down-regulated by various abiotic stresses (*Wen et al., 2014*). In this study, the detected *BvWRKY* genes showed a variety of expression patterns in shoots and roots of sugar beet seedlings under various concentrations of $NaHCO_3$ (Fig. 8). When plants are exposed to alkaline stress, the expression levels of *BvWRKY10*, *-42*, and *-47* in shoots slowly raised at 15 and 25 mM, then reached a peak value at 50 mM, and then either maintained a higher level (*BvWRKY10* and *-42*) or showed a significant decrease (*BvWRKY47*) at 100 mM (Figs. 8B, 8F and 8H). Similarly, *BvWRK16* and *-42* in roots displayed the same expression pattern. It was also observed that *BvWRKY3* and *-51* in shoots were unchanged and down-regulated by all the concentrations of alkaline (Figs. 8A and 8I). However, four genes (*BvWRKY22* and *-41* in shoots, or *BvWRKY44* and *-51* in roots) responded to alkaline stress in an irregular pattern (going down, up and then down, or doing up, down and then up with the increase of alkaline concentrations) (Figs. 8D, 8E, 8G, and 8I). This expression pattern might be attributed to the variety of the *BvWRKY* genes transcription. Importantly, the mRNA levels of *BvWRKY10* in shoots and *BvWRKY16* in roots were 2.2- and 12.6-fold higher at 50 mM $NaHCO_3$ than those at control condition (0 mM) (Figs. 8B and 8C). Additionally, GO analysis showed that most of the *BwWRKY* genes are classified into functional groups of "sequence-specific DNA binding (GO:0043565)" and "transcription factor activity (GO:0003700)" (Fig. 6 and Table S2). These results implied that these *BvWRKY* genes might positively respond to alkaline stress via binding to W-boxes (TTGACC/T) of in the promoter regions of downstream genes related to alkaline tolerance (*Rushton et al., 2010*). However, their detailed functions in alkaline tolerance of sugar beet need to be further addressed by using chromatin immunoprecipitation (ChIP), RNA interference, gene overexpression or editing methods. Overall, our results provide useful information for studying the effects of the *BvWRKY* genes in sugar beet under alkaline stress.

## CONCLUSION

In the present study, a total of 58 putative *BvWRKY* genes were identified in the sugar beet genome. Based on the primary amino acid sequence, the *BvWRKY* genes are classified into three major groups I, II, and III, each with 11, 40, and seven genes, respectively. *BvWRKY* genes of group II are further divided into five distinct subgroups IIa, b, c, d, and e, containing three, seven, 15, seven, and eight members, respectively. All the identified BvWRKY proteins have one or two conserved WRKY domains and one zinc-finger structure. The number of introns in the *BvWRKY* genes range from one to five, with the majority of *BvWRKYs* containing three exons. Furthermore, the detected *BvWRKY* genes showed a variety of expression patterns in shoots and roots of seedling exposed to alkaline stress. Notably, the transcript levels of *BvWRKY10* in shoots and *BvWRKY16* in roots are significantly higher at 50 mM alkaline than those at control condition. This study provides a wide identification of the *BvWRKY* genes, and would be helpful for the improvement of alkaline tolerance in sugar beet by genetic engineering.

## ACKNOWLEDGEMENTS

We would like to thank Mr. Shengfu Duan for kindly providing seeds of sugar beet. We are very grateful to Prof. Jin-Lin Zhang from Lanzhou University for critically reviewing the manuscript and for valuable suggestions. We are also thankful to two reviewers for their constructive comments on the manuscript.

### Funding

This work was supported by the National Natural Science Foundation of China (No. 31860404) and the Natural Science Foundation of Gansu Province (No. 18JR3RA152). The funders had no role in study design, data collection and analysis, decision to publish, or preparation of the manuscript.

### Grant Disclosures

The following grant information was disclosed by the authors:
National Natural Science Foundation of China: 31860404.
Natural Science Foundation of Gansu Province: 18JR3RA152.

### Competing Interests

The authors declare there are no competing interests.

### Author Contributions

- Guo-Qiang Wu conceived and designed the experiments, prepared figures and/or tables, authored or reviewed drafts of the paper, approved the final draft.
- Zhi-Qiang Li performed the experiments, analyzed the data, prepared figures and/or tables, approved the final draft.
- Han Cao and Jin-Long Wang performed the experiments, approved the final draft.

### Data Availability

The following information was supplied regarding data availability: Raw data is available in the Supplemental File.

### Supplemental Information

Supplemental information for this article can be found online at http://dx.doi.org/10.7717/peerj.7817#supplemental-information.

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
