# Peer review of "Genome-wide identification and expression analysis of the WRKY genes in sugar beet (Beta vulgaris L.) under alkaline stress"

_PeerJ, doi:10.7717/peerj.7817_

## Round 0.1 · original submission · Major Revisions

Both reviewers gave positive feedback on your work concerning the scientific contribution of your study. However, they also made several comments for discussing and presenting your results, and you should take them into account.

·

Basic reporting

no comments

Experimental design

Phenotypic data lacking?

Validity of the findings

Why only five genes selected for validation?

Additional comments

Authors; Wu et al. in this study identified 58 WRKY TFs from sugarbeet genome using insilico analysis and did analyze for its chromosomal positions, gene structure, conserved motifs etc. Further, they validated a few genes using qRTPCR. Authors have generated new information which was not reported earlier. I am just wondering why the author did not record the phenotypic data of samples used for alkaline stress? why the author choose only two tissues? root and shoot why not other parts and stage data for correlations with transcripts at what stage is express more?
i have few corrections and suggestions as listed below:
1. Lines 60-63 needs to be rewritten for instance say" adversely environments'?
2. Line no. 76 "different kind of plants"? what do you mean by this
3.Line 97. what is "lower water loss"?
4. Lines 231-233 " make a new phrase
5. Line 250-252 " for validation of few genes using qRTPCR what is the basis of selection is not clear?
6. Line 338-339 " incomplete sentence?

Reviewer 2 ·

Basic reporting

Wu et al., reported the identification and phylogenetic classification of WRKY genes in sugar beet and demonstrated their potential involvement in alkaline stress. The language of this manuscript is okay. However, the authors should take more efforts in improving their experimental design and elaborating their results and providing more convincing evidence on the role of WRKY genes in alkaline stress before publication. Furthermore, they need to improve the quality and resolution of all figures.

Experimental design

a.The authors should include a step of motif scan of WRKY domain in the ‘identification of WRKY genes’. Although it turned out that all genes contain WRKY motifs in their ‘conserved motif analysis’, it would be more logical if they perform that earlier. Genes without WRKY domain should be excluded from their analysis. Meanwhile, they need to explain what was used for the identification of WRKY genes, was it based on the domain sequence from Pfam (L142)?
b. The authors stated that they removed overlapping sequences manually (L190). This step raises a lot of doubts. First, how did the authors define ‘overlapping’?. Were those sequences from different genome loci? Second, it is possible that the authors removed WRKY genes that were caused by genome duplication (segmental duplication). It is not reasonable to remove those sequences. In addition, an assay for segmental duplication is highly recommended in this case since they mentioned that in the discussion (L284).
c.The authors mentioned that they named WRKY genes according to their order in the sugar beet genome sequence. However, they were ordered from a mixture of different chromosomes in Table 2. Please explain this.
d.’WRKYs are classified into 3 major groups ..based on their putative WRKY domains’. The grouping of phylogenetic trees should be determined by how they clustered rather than their domains. If the authors would like to group them by domains, what is the necessity of constructing a tree? In addition, they did not mention any result of domain scanning ahead of this. It seems that they used their later results in this analysis, which is not logical.

Validity of the findings

a.The authors said ‘similar observations were made in sesame’ (L198). First, I did not find any similarity between sugar beet and sesame. Second, I am not sure why this statement is informative to the audience.
b.The authors mentioned that WRKY genes are evenly distributed throughout the genome, which is not true (L212). In addition, the statement of ‘the enrichment of WRKY genes on chromosome 5’ is not convincing, given that there are 9 genes on chr 2 and chr2 is even shorter. Meanwhile, the scale on the side of Figure 2 is not correct, at least not proportional. Also what they marked in red does not fully match with the text. Finally, please explain why 3 genes are located on the chromosome unknown. Is that because they are located on unassembled scaffolds?
c. The authors claimed that ‘ WRKY genes in the same cluster share similar motif compositions’ (L239). However, I could not draw the same conclusion from Figure 3 since it is not organized according to their cluster. The authors should modify accordingly.
d. The authors’ statement that ‘WRKY genes within the same group share a similar structure’ is also not well-supported by their description (L248).
e. The authors did not mention why they only perform qPCR on 9 genes and how were those genes selected. Are those genes the only ones with detectable expression levels? If yes, they need to mention that. If not, they need to either provide a reason why group III was not included or perform more qPCR on the remainder of genes. And there are lots of problems in their descriptions. For example, they claimed that WRKY1,4,9,17,31 are all induced by the stress treatment. However, in some conditions, they are repressed or unchanged. I would recommend to rewrite this paragraph. They also need to explain why most genes respond to stress in an irregular pattern (going up and then down, or doing up ,down, up and then down with the increase of salt concentration). Furthermore, since this paper emphasizes on the role of WRKY in alkaline stress. The authors need to perform more analyses (e.g. prediction of gene functions using GO terms) to support their conclusion. So far, it is not convincing.
f. The authors hypothesis that ‘WRKY31 is involved in the regulation of organic acid genes’ is reckless and further evidence is needed for supporting this hypothesis.

Additional comments

a.It would be better if the authors could mention TaWRKY1 is from wheat (L93).
b.Table 1 should be supplemental.
c.L304, should be Table 2.
d.Please explain what circles and squares in Figure 1 refer to and why some genes are labeled in red. The three groups should also be labeled.
e.Figure 4, does ‘upstream and downstream’ mean UTR?

---

## Round 0.2 · Minor Revisions

Your manuscript has been substantially improved and you provided adequate responses to the reviewers' doubts. Could you please address the few remaining minor comments?

Reviewer 2 ·

Basic reporting

The authors have addressed most of my concerns. However, there are a few new points need to be clarified.
1. It seems that the clustering of WRKYs in Figure 1 into 3 groups is not based solely on phylogenetic analysis. The authors should mention where the grouping of I, II, III and IIa etc. is derived from. Presumably from Arabidopsis studies? (L325).
2. Figure S2, motif 3 is not a WRKY domain?
3. For the GO enrichment analysis, the authors should how the fold enrichment too.
4. For the 3D structure prediction, I am not sure how the result would facilitate the understanding of the functional mechanism of BvWRKY proteins.
5. L186, should be TAIR.
6. L312, the authors stated that 58 BvWRKY proteins had one or two conserved WRKYGQK domains. However, in Fig S1 it looks like there is only one domain. And if there were two, which one was used for alignment and for the phylogenetic analysis?

Experimental design

no comment

Validity of the findings

no comment

Additional comments

no comment

---

## Round 0.3 · Minor Revisions

Thank you very much for addressing all additional comments in your revised manuscript. However, our Section editor made the following observation:

> The GO annotations are only illustrated in Fig 6; there needs to be a table with the GO: annotation identifier connecting it to the sequences used. This is considered minor at this point, but would like to see an extra column in something like table S2 indicating the assigned GO: terms to the sequences as illustrated in Figure 6. Perhaps this should remain as a ‘minor revision until this can be added. I did see that the sequences were added to Table S1.>

Could you please add the GO annotation in your main manuscript?

---

## Round 0.4 · accepted · Accept

Thank you very much for addressing the additional comments on your manuscript.